# Spatial Characterization of Woody Species Diversity in Tropical Savannas Using GEDI and Optical Data

**DOI:** 10.3390/s25020308

**Published:** 2025-01-07

**Authors:** Franciel Eduardo Rex, Carlos Alberto Silva, Eben North Broadbent, Ana Paula Dalla Corte, Rodrigo Leite, Andrew Hudak, Caio Hamamura, Hooman Latifi, Jingfeng Xiao, Jeff W. Atkins, Cibele Amaral, Ernandes Macedo da Cunha Neto, Adrian Cardil, Angelica M. Almeyda Zambrano, Veraldo Liesenberg, Jingjing Liang, Danilo Roberti Alves De Almeida, Carine Klauberg

**Affiliations:** 1Department of Forestry Engineering, Federal University of Paraná-UFPR, Curitiba 80050-380, PR, Brazil; anapaulacorte@gmail.com (A.P.D.C.); netomacedo878@gmail.com (E.M.d.C.N.); 2Forest Biometrics and Remote Sensing Laboratory (Silva Lab), School of Forest, Fisheries, and Geomatics Sciences, University of Florida, P.O. Box 110410, Gainesville, FL 32611, USA; c.silva@ufl.edu (C.A.S.); carine.klaubergs@ufl.edu (C.K.); 3Spatial Ecology and Conservation (SPEC) Laboratory, School of Forest, Fisheries, and Geomatics Sciences, University of Florida, Gainesville, FL 32611, USA; eben@ufl.edu; 4NASA Postdoctoral Program Fellow, Goddard Space Flight Center, Greenbelt, MD 20771, USA; rodrigo.vieiraleite@nasa.gov; 5Biospheric Sciences Laboratory, Code 618, NASA Goddard Space Flight Center, Greenbelt, MD 20771, USA; 6US Department of Agriculture, Forest Service, Rocky Mountain Research Station, 1221 South Main Street, Moscow, ID 83843, USA; andrew.hudak@usda.gov; 7Federal Institute of Education, Science and Technology of São Paulo-IFSP, Cubatão 11533-160, SP, Brazil; hamamura.caio@ifsp.edu.br; 8Faculty of Geodesy and Geomatics Engineering, K.N. Toosi University of Technology, P.O. Box 15875-4416, Tehran 15418-49611, Iran; hooman.latifi@kntu.ac.ir; 9Department of Remote Sensing, University of Würzburg, Oswald Külpe Weg 86, 97074 Würzburg, Germany; 10Earth Systems Research Center, Institute for the Study of Earth, Oceans, and Space, University of New Hampshire, Durham, NH 03824, USA; j.xiao@unh.edu; 11USDA Forest Service, Southern Research Station, P.O. Box 400, New Ellenton, SC 29809, USA; jwatkins6@vcu.edu; 12Environmental Data Science Innovation and Inclusion Lab (ESIIL), Cooperative Institute for Research in Environmental Sciences, University of Colorado Boulder, Boulder, CO 80303, USA; cibele.amaral@colorado.edu; 13Earth Laboratory, Cooperative Institute for Research in Environmental Sciences, University of Colorado Boulder, Boulder, CO 80303, USA; 14Technosylva Inc., La Jolla, CA 92037, USA; adriancardil@gmail.com; 15Department of Crop and Forest Sciences, University of Lleida, 25001 Lleida, Spain; 16Joint Research Unit CTFC-AGROTECNIO-CERCA, 25198 Solsona, Spain; 17AX Spatial Ecology and Conservation (SPEC) Lab, Center for Latin American Studies, University of Florida, Gainesville, FL 32611, USA; aalmeyda@ufl.edu; 18Department of Forest Engineering, College of Agriculture and Veterinary, Santa Catarina State University (UDESC), Lages 88520-000, SC, Brazil; veraldo@gmail.com; 19Forest Advance Computing and Artificial Intelligence Laboratory, Department of Forestry and Natural Resources, Purdue University, West Lafayette, IN 47907, USA; alpenbering@gmail.com; 20Department of Forest Sciences, “Luiz de Queiroz” College of Agriculture, University of São Paulo (USP/ESALQ), Piracicaba 13418-900, SP, Brazil; daniloflorestas@gmail.com

**Keywords:** Cerrado, alpha diversity, GEDI, LiDAR, imagery, modeling

## Abstract

Developing the capacity to monitor species diversity worldwide is of great importance in halting biodiversity loss. To this end, remote sensing plays a unique role. In this study, we evaluate the potential of Global Ecosystem Dynamics Investigation (GEDI) data, combined with conventional satellite optical imagery and climate reanalysis data, to predict in situ alpha diversity (Species richness, Simpson index, and Shannon index) among tree species. Data from Sentinel-2 optical imagery, ERA-5 climate data, SRTM-DEM imagery, and simulated GEDI data were selected for the characterization of diversity in four study areas. The integration of ancillary data can improve biodiversity metrics predictions. Random Forest (RF) regression models were suitable for estimating tree species diversity indices from remote sensing variables. From these models, we generated diversity index maps for the entire Cerrado using all GEDI data available in orbit. For all models, the structural metric Foliage Height Diversity (FHD) was selected; the Renormalized Difference Vegetation Index (RDVI) was also selected in all species diversity models. For the Shannon model, two GEDI variables were selected. Overall, the models indicated performances for species diversity ranging from (R^2^ = 0.24 to 0.56). In terms of RMSE%, the Shannon model had the lowest value among the diversity indices (31.98%). Our results suggested that the developed models are valuable tools for assessing species diversity in tropical savanna ecosystems, although each model can be chosen based on the objectives of a given study, the target amount of performance/error, and the availability of data.

## 1. Introduction

The Brazilian savanna, also known as the Cerrado, is home to over 12,400 plant species [1] and accounts for one-third of Brazil’s biodiversity, with a high rate of endemism, making it the most biodiverse savanna globally [2]. The Cerrado is considered the largest savanna in South America and is the continent’s second-largest ecosystem after the Amazon, and so it plays a crucial role in global ecological processes [2]. However, this critical biome faces significant degradation and anthropogenic pressures. As of 2022, approximately half of the Cerrado had been converted into pastureland (51.6%) or agricultural fields (26%) [3], with only 8.2% falling under formal protection in indigenous reserves or parks [4]. Historically, forest cover change assessments in South America have largely centered on tropical rainforests, often neglecting regions with shorter humid seasons like the Cerrado [5,6]. As a consequence of vegetation cover changes, tree species biodiversity declined substantially [7], resulting in the loss of biodiversity and important ecosystem services, such as nutrient and water cycle regulation, soil protection, and food and wood provision [8,9].

The spatial monitoring of ecosystem structures and diversity metrics is therefore crucial. Innovative approaches are essential in providing data that can effectively guide conservation policies, inform climate change mitigation strategies [10,11], and support the sustainable management of natural resources. However, the conservation of tropical savannas is challenged by the lack of detailed information on species diversity on a regional scale.

The conservation of biodiversity in threatened ecosystems like the Cerrado requires approaches that integrate structural and spectral data and so can overcome the limitations of traditional methods. Floristic inventories, although detailed, are limited in scale and incur high costs, and present challenges such as time consumption and difficulties in standardizing reproducible procedures [12]. Optical imagery often faces signal saturation in high-biomass areas, which hinders the accurate discrimination of vegetation structure and composition. Remote sensing, on the other hand, offers a transformative alternative, enabling the cost-effective mapping of species diversity across extensive areas [13,14,15]. Compared to traditional field methods, remote sensing provides a scalable solution for biodiversity monitoring and could be critical in implementing effective conservation strategies and mitigating biodiversity losses [16].

Several studies have explored species diversity in savanna ecosystems using remote sensing [17,18,19,20,21]. However, research focused on the Cerrado remains scarce and predominantly relies on passive optical data, which have limitations due to signal saturation in areas with higher biomass [20]. In addition, most studies that used multispectral information explored only the Normalized Difference Vegetation Index (NDVI) to assess tree species diversity [17,19]. Other broadband VIs have been used, though more rarely [22]. Despite being widely used in vegetation and remote sensing studies, NDVI saturation capacity affects the discrimination of vegetation structure and composition as leaf area index peaks [23]. However, conventional passive optical imagery from air- and space-borne platforms have limited applicability, as these data are not directly sensitive to vertical vegetation structure [24].

Active remote sensing, particularly LiDAR technology, has proven effective in accurately measuring spatial and vertical vegetation structures [10,25,26]. Airborne LiDAR has yielded highly accurate results for assessing species diversity [13,27,28,29,30]. However, most of these studies used airborne LiDAR data, which is limited by high survey costs and low area coverage. Spaceborne LiDAR platforms like GEDI overcome these limitations, delivering global-scale data on forest structure and biodiversity potential. Combining spaceborne LiDAR from GEDI with optical data offers a unique opportunity to capture both the vertical structure and spectral composition of vegetation. This approach is particularly relevant for the Cerrado, where signal saturation in vegetation indices limits the use of optical data alone.

The GEDI mission, launched by NASA in 2018, represents a milestone in global observation, offering unprecedented opportunities for large-scale ecological and biodiversity studies. Operating from the International Space Station (ISS), GEDI provides high-resolution data on vegetation structure, with a focus on forest biomass and diversity [31]. Previous studies have demonstrated the utility of GEDI for assessing biodiversity in savanna ecosystems, explaining up to 71% of species diversity variance in African savannas [32]. However, whether such methods can be reliably applied to the Brazilian Cerrado remains an open question, particularly given that biome’s unique environmental and ecological characteristics.

Furthermore, to the best of our knowledge, no study has yet proposed to assess the combined contribution of variables from different sensors, such as spectral vegetation indices (VIs), canopy structure, topography, and climate data. Factors such as altitude, aspect, precipitation, and temperature all play a decisive role in the distribution of vegetation communities [33]; however, these factors have rarely been taken into account in assessing species diversity. Moreover, while previous studies have predominantly focused on African savannas or relied on limited optical data for biodiversity assessments, this research provides a novel approach by integrating GEDI LiDAR data with spectral, climatic, and topographic variables in order to model biodiversity indices in the Cerrado. This innovative methodology not only addresses the challenges of signal saturation in high-biomass regions but, also, offers insights into how environmental variables interact with vegetation structure and composition on a continental scale, filling a critical gap in the literature on tropical savanna ecosystems.

The main objective of this study is to develop and evaluate multi-source predictive models that combine GEDI structural data with optical, climate, and topographic information to estimate three biodiversity indices across the Cerrado: Shannon, Simpson, and species richness. This study innovates by integrating GEDI structural data with climatic, topographic, and spectral variables to model biodiversity indices on a continental scale. This multi-source approach allows the exploration of environmental variable interactions with vegetation structure and composition metrics. Additionally, we also aim to characterize large-scale species diversity across the entire Cerrado (i.e., 1.9 million km^2^) by applying calibrated Random Forest (RF) models to multi-source data, then aggregating footprint levels, then estimating three species diversity indices to a 1 km resolution grid across the biome.

## 2. Materials and Methods

### 2.1. Study Area

The study area spans the states of Minas Gerais and Goiás, covering the savanna forest strips in the Cerrado, Brazil (Figure 1). The area is divided into two land management regimes with different land-use practices: communal areas (University Federal of São João del-Rei-UFSJ-1 km^2^); and protected areas (Serra do Cipó National Park-CNPK-316 km^2^, Chapada dos Veadeiros National Park-CVNPK-2.406 km^2^, Paraopeba National Forest-PNF-2 km^2^).

The PNF, UFSJ, and CNPK study sites are located in the southeast portion of the Cerrado, in the state of Minas Gerais; the CVNPK is located in the central portion of the Cerrado (Figure 1). Each site is characterized by different climatic, topographic, and water regime conditions; there are also diverse vegetation formations present in the Cerrado, which must be considered. More detailed information can be found in [34]. Overall, open grasslands are characterized by the predominance of herbaceous species and some shrubs, and an absence of trees. Savanna formations refer to areas with trees and shrubs scattered over a stratum of grasses and herbs, with an absence of continuous tree canopies. Forest formations are areas with a predominance of tree species with either a continuous or discontinuous canopy [35].

The landscape of the CVNPK (13°51′–14°10′S, 47°25′–42′W) consists of a mosaic of diverse vegetation types [35]. Lower elevations are predominantly covered by forest formations, while higher elevations feature montane savannas. Wet and dry grasslands, as well as savannas, extend between stream corridors and make up the majority of the landscape [36]. At the northwest edge of the park, dry, deciduous forests are present, while riparian evergreen forests dominate the southwest edge [37]. Overall, approximately 77% of the CVNPK is made up of savanna formations, with around 10% consisting of forest fragments [38]. The CNPK displays a range of vegetation physiognomies, from open grasslands (“campo limpo”) below 1000 m altitude to savanna formations with varying degrees of woody cover (“campo sujo”, “campo cerrado”, and “cerrado sensu stricto”) and forest formations (“cerradão”), all classified as part of the broader Cerrado sensu lato [39]. Additionally, the rupestrian grasslands are found at elevations above 1000 m [40].

For the characterization of the vegetation types in the Cerrado, herein, we follow the definitions proposed by [35], which subdivide the Cerrado into open grasslands (campo sujo, rupestrian grasslands, and campo limpo), savanna formations (campo cerrado, cerrado sensu stricto, palm grove, and veredas, classified as wet savanna), and forest formations (riparian forest, gallery forest, dry forest, and cerradão). In addition, the Cerrado physiognomic gradient is related to environmental factors and is maintained by spatially and temporally dynamic disturbance regimes, both natural and anthropogenic [41].

### 2.2. Field and Forest Diversity Data

A total of 50 square plots, each measuring 900 m^2^ (30 × 30 m), were measured between June and July of 2019 for this study, representing 0.0016% of the total area of the study sites. Plot corners were registered using a Differential Global Navigation Satellite System (DGNSS). Each tree was taxonomically identified, and its height (ht, in m) and diameter at breast height (dbh, in cm) were measured using a clinometer and diameter tape, respectively. To ensure reproducibility, three diversity indices (species richness, Simpson index, and Shannon index) were calculated for each plot (Table 1).

Species richness refers to the total number of different species in a sampling unit and provides a direct measure of biodiversity [29]. Other diversity indices combine two attributes within a community: species richness and equability [42,43]. Equability refers to how similarly species are represented in the community. If all species have the same representativeness (or importance [42]), the equability will be at a maximum. Most diversity indices are said to be non-parametric as they do not depend on the parameters of a distribution. They usually consist of simple mathematical expressions involving the relative abundance of each species in the sample [44].

The Shannon–Wiener diversity index, represented by H′, is calculated based on the number of individuals in each species and the total number of individuals sampled. As one of the most frequently used diversity indices, it is sensitive both to species’ rarity and abundance, and it has been used in various studies as a measure of alpha diversity [19,29]. The Simpson index is also a widely used measure of species diversity [45,46], which takes into account the abundance and number of species present in an area and estimates the probability that two individuals chosen at random belong to the same species.

### 2.3. Remote Sensing Data

This study integrated four data sources: Sentinel-2 optical imagery, ERA-5 climate data, SRTM-DEM, and simulated GEDI data. Data collection and processing for the first three sources were conducted via Google Earth Engine (GEE); simulated GEDI data were generated using the rGEDI package in R [47].

#### 2.3.1. UAV-LiDAR GatorEye

We simulated GEDI data from the UAV-lidar 3D point cloud for calibrating species diversity models in order to avoid the geolocation errors accrued by GEDI (~10–20 m) and due to the fact that GEDI orbits likely did not overlay our field plots. The GatorEye UAV-LiDAR system [48] was selected to scan our study sites for two weeks in July 2019, almost simultaneously with field data collection. The GatorEye used a DJI M600 Pro flight platform mounted with a Phoenix Scout Ultra core to integrate LiDAR (VLP16) with an inertial motion unit (Novatel STIM 300, NovAtel Inc., Calgary, AB, Canada), and centimeter accuracy differential GNSS system. A complete description of the GatorEye system can be found in a recent study [34]; data are also available on the GatorEye website (www.gatoreye.org, accessed on 17 December 2024). For further information, the reader is referred to [48,49]. The autonomous flight was programmed to survey at a mean speed of 14 m s^−1^ at 100 m above ground level (AGL), with flight lines spaced 100 m apart. In total, across the four study sites, we flew more than 600 km of flight lines, with a lidar swatch coverage of 1854 hectares, which, to our knowledge, as of the flight date, is the largest area of UAV-lidar used in a publication and the only one in this ecosystem. This mapped area represents approximately 7% of the total study area. Validation of the simulated GEDI data was performed by comparing extracted canopy metrics with field-measured plot characteristics. The final merged point clouds were about 100 GB in total size and had a very high density of approximately 450 points m^−2^ across all study sites [39]. In this study, processing UAV-LiDAR 3D point cloud data involved applying the GatorEye Multi-Scalar Post-Processing Workflow (as described in [48]), aligning flight paths, and clipping point clouds to fit the field plots for simulating GEDI data [50].

#### 2.3.2. NASA’S GEDI

GEDI data from the UAV-LiDAR 3D point cloud were simulated using the approach outlined in [50]. The GEDI pre-launch planning phase involved creating a simulator capable of replicating the characteristics of in-orbit GEDI data [51]. The simulation process involves converting discrete-return lidar point clouds into full-waveform signals within GEDI-sized footprints and incorporating anticipated noise levels from the GEDI instrument [52]. The signal-to-noise ratio (SNR) in the in-orbit GEDI data is influenced by factors such as laser type (power or coverage), time of acquisition (day or night), canopy density, and atmospheric conditions [51,53]. This simulator maintains consistency across flight characteristics, particularly for high-density LiDAR point clouds, like those utilized in this study, which ensures that models can be reliably transferred to in-orbit GEDI data. A comprehensive explanation and validation of the GEDI simulator can be found in [51].

GEDI waveforms were simulated using the gediWFSimulator tool from the rGEDI package, which introduces realistic noise levels to mimic in-orbit GEDI data. Metrics such as canopy cover (COV), foliage height diversity (FHD), and relative heights (RH10, RH25, and RH50, etc.) were calculated to match GEDI Level 2A and 2B products [47,54]. Although the simulation accurately replicates GEDI’s operational metrics, it is important to acknowledge potential uncertainties associated with these data, including noise levels and discrepancies between simulated and actual GEDI acquisitions. Realistic noise was introduced based on a beam sensitivity of 0.98, representing canopy cover where the ground is detected 90% of the time with a 5% false positive probability, following [51]. This was achieved with a link margin of 4.956 under 95% canopy cover, corresponding to noise levels for the power beam collecting data at night [55]. For ground detection and metric calculations, waveforms were denoised and smoothed by setting the noise threshold to the mean plus three standard deviations, with a smoothing width of 0.5 m applied post-denoising [56,57].

#### 2.3.3. Sentinel-2 MSI

Sentinel-2 offers multi-spectral data that include four bands with 10 m spatial resolution, six bands at 20 m, three bands at 60 m, and three quality assessment (QA) bands, with QA60 serving as a bitmask band containing cloud mask information [58]. In this study, we used Sentinel-2 level 2A surface reflectance products available in the Google Earth Engine (Dataset ID: ee.ImageCollection (“COPERNICUS/S2_SR”)). We selected images from 1 May 2019 to 31 August 2019, with less than 30% cloud cover based on the “CLOUDY_PIXEL_PERCENTAGE” attribute, and combined them to reduce cloud interference. Then, we computed the following vegetation indices (VIs): Normalized Difference Vegetation Index (NDVI), Enhanced Vegetation Index (EVI), Soil-Adjusted Vegetation Index (SAVI), Renormalized Difference Vegetation Index (RDVI), and Simple Ratio Index (SRI).

On GEE, we applied a compound median-reducing function to calculate the median value of each image collection from May to August, i.e., the Cerrado’s dry season. The median-reducing function removes clouds, which have high values, and shadows, which have low values, from the image. The output composite value is the median in each band over time. Then, we applied a clip function to group the image collections in the study region, then calculated the indices using equations in Table 2.

The NDVI is one of the most commonly used remotely sensed spectral vegetation indices and is calculated from reflectance in the near-infrared and red portions of the electromagnetic spectrum [59]. The Enhanced Vegetation Index (EVI) was proposed by Liu and Huete [60] to compensate for the limitations of the NDVI regarding soil background and atmospheric interference. Generally, NDVI is responsive to chlorophyll content and other pigments that absorb solar radiation in the red portion of the electromagnetic spectrum. In contrast, EVI is additionally sensitive to variations in canopy structure, including Leaf Area Index (LAI), plant physiognomy, and canopy volume, due to the incorporation of blue band information [61,62].

The RDVI was proposed to combine the advantages of the Difference Vegetation Index (DVI = NIR − Red; [63]) and the NDVI for low and high LAI values, respectively. The RDVI was proposed to minimize saturation effect. The Soil-Adjusted Vegetation Index (SAVI; [62]) was proposed to account for changes in soil optical properties. The SAVI includes a canopy background adjustment factor L. Finally, the Simple Ratio (SR) index was selected as it is one of the most commonly used vegetation indices [64,65]. It provides unique information that is not available in any single band. It is used for discriminating between soil and vegetation in the study region [66].

#### 2.3.4. Ancillary Data

ERA-5 climate data and SRTM-DEM provided essential ancillary variables for diversity modeling. These datasets complemented remote sensing data by accounting for climate and topographic variations across the study sites. We used the ERA-5 [67] and Digital Elevation Model (DEM), obtained from Shuttle Radar Topography Mission (SRTM) data [68,69], as ancillary data for forest diversity modeling. The ERA-5 (fifth generation) is the latest climate reanalysis model produced by the ECMWF (European Centre for Medium-Range Weather Forecasts) and the Copernicus Climate Change Service [70], with a spatial resolution of 31 km. This dataset is freely available and offers a detailed overview of the atmosphere. In addition, the ERA-5 is part of GEE’s datasets, consisting of air temperature as a monthly average at 2 m height, with data available from 1979 to present. With this method, we selected the products “mean_2m_air_temperature”, which corresponds to the average air temperature at 2 m height (monthly average), as well as “total_precipitation”, which refers to total precipitation (monthly sums). These were chosen as the variables to represent the temperature and precipitation of the study areas.

SRTM data, measured and released by NASA and the National Surveying and Mapping Bureau of the US Department of Defense, cover 80% of global land surface [71]. Our research utilized the SRTM Version 3 (V3) product, available from NASA’s Jet Propulsion Laboratory (JPL), at a 1 arc-second resolution (around 30 m). Using the code provided on the GEE platform (https://developers.google.com/earth-engine/datasets/catalog/USGS_SRTMGL1_003, accessed on 25 December 2024) we applied functions to directly extract aspect, slope, and elevation for our study areas. The remote sensing candidate metrics used in this study are summarized in Table 3. These metrics include data from multiple sources, such as GEDI, Sentinel 2, ERA 5, and SRTM, which were selected to enhance the modeling of forest species diversity.

### 2.4. Feature Selection

Feature selection is a crucial step in machine learning with two primary purposes: (1) reducing the number of features and dimensions, thereby improving model generalization and minimizing overfitting; and (2) clarifying the relationships between features and eigenvalues. The VSURF algorithm was applied to the original dataset to identify the optimal number of variables. This wrapper-based algorithm uses Random Forest (RF) as the base classifier [72]. Initially, feature variables are ranked by an importance measure, and those with lower scores are removed to reduce feature count and enhance model accuracy [73]. The process ultimately produces a ranked list of only the most significant features.

Then, we applied an RF regression [74] for predicting diversity indices. RF has been widely used in forest modeling based on earth observation data [26,75,76], due to its non-parametric nature and its ability to deal with dimensionality, multicollinearity, and overfitting [77,78]. In RF modeling, two parameters are required to construct the decision trees. The first one is the number of decision trees that should be generated. The second is the number of variables that need to be selected for the greatest split when the trees become larger over a period of time (k predictor). The RF algorithm was implemented in R using the randomForest package [79].

The bootstrapping approach was applied in order to assess the precision and accuracy of the models. First, we completed 1000 random permutations of the original data, then split the data to a training set and test data set. Two-thirds of the data were used to train the models; the remaining data were used to assess the predictive ability of the models. The strength of the relationship was assessed using the coefficient of determination (R^2^); the performance of the model was assessed using root mean square error (RMSE) and bias. All diversity measures and regressions were calculated using RStudio v1.4.17 [47]. Figure 2 shows the bootstrapping process for modeling diversity indices in the Brazilian Cerrado.

The diversity models were based on multi-source data from remote sensing and in situ measurements. An overview of the methodology is illustrated in Figure 3. The approach was divided into three phases: (1) Field surveys and calculating diversity indices; (2) remote sensing and processing data collection; and (3) development of predictive models and validation.

### 2.5. Development and Validation of Predictive Models

In order to characterize the Cerrado species diversity indices, we carried out a series of sequential steps. First, GEDI Level 2A and 2B (version 2) data [80,81], collected between 18 April 2019 and 1 March 2023, were downloaded for the entire vegetated area of the Cerrado. GEDI orbits intersecting the Cerrado boundaries were identified and downloaded from the rGEDI package using the gedifinder and gediDownload functions [60]. Next, the GEDI footprints were masked to the vegetated area of the Cerrado based on the MapBiomas land cover classification for the corresponding year of data collection [82]. This procedure ensured that only pixels classified as forest, savanna, and grassland vegetation were considered in the subsequent steps. The data were then integrated into a unified environment. In the Google Earth Engine (GEE), optical data layers (Sentinel-2), elevation data (SRTM), and climate data (ERA-5) were stacked using the layerstack function. The combined data were imported into the R environment, where they were integrated with the GEDI footprints. In this way, each GEDI footprint included not only the previously selected structural metrics but, also, the values of the optical, climate, and elevation variables. The footprint-level metrics were extracted using the getLevel2A and getLevel2B functions in the rGEDI package. These metrics include key structural variables such as canopy height, canopy cover, and the vertical profiles of vegetation structures. To ensure the use of only high-quality GEDI data, the metrics were filtered using the “quality_flag = 1” parameter. This filter guarantees that the selected data meet strict quality criteria, including waveform shot energy, signal sensitivity (<0.9 over land), amplitude, and real-time surface tracking quality [82,83]. These criteria reduce uncertainties stemming from sensor noise or atmospheric interference, ensuring that the GEDI measurements are both reliable and accurate. Once high-quality data were selected and pre-processed, the diversity index models were applied at the GEDI footprint scale. Each footprint, with an approximate diameter of 25 m, represents a localized sampling point where structural vegetation metrics are combined with additional environmental datasets. The integration of GEDI data with optical (Sentinel-2), climate (ERA-5), and elevation (SRTM) variables allowed for a robust assessment of species diversity across the Cerrado biome. This integration process involved stacking optical and ancillary data within a Google Earth Engine (GEE) environment and, subsequently, linking these datasets to the GEDI footprints within the R environment. By spatially aligning these inputs, each GEDI footprint became enriched with a suite of predictor variables that enhance the accuracy of the diversity index estimates. The diversity index estimates at the footprint level were then spatially aggregated into 1 km^2^ grid cells. This spatial resolution was chosen to ensure compatibility with planned GEDI gridded products [84] and to meet the requirements of global biomass mapping initiatives [85], which rely on consistent and standardized data at this scale. The aggregation process involved averaging the diversity index estimates across all GEDI footprints within each grid cell, reducing spatial variability while providing a generalized representation of diversity patterns at a landscape scale. Finally, the uncertainty of the diversity index estimates was quantified for each 1 km^2^ grid cell. This step was essential in evaluating the robustness and reliability of the predictions. The uncertainty was calculated by accounting for the variability of footprints within each cell, the uncertainty associated with the Random Forest (RF) algorithm, and the residual lack of model fit. The variability among footprints within a cell reflects the heterogeneity in vegetation structure and composition; the RF uncertainty captures any errors inherent in the predictive model and the limitations of the training dataset. The residual lack of fit indicates deviations between predicted and observed values, particularly in complex or poorly represented areas. The calculation of uncertainty followed the methodology proposed by [50], ensuring consistency with previous studies and providing a clear framework for interpreting the confidence of the diversity index predictions. By incorporating multiple sources of uncertainty, the resulting diversity index maps deliver not only estimates of species diversity but, also, a robust evaluation of their associated confidence, which enhances their value for ecological monitoring and conservation planning.

## 3. Results

### 3.1. Species Diversity Indices and Remote Sensing Metrics

Different types of species diversity indices showed similar patterns in variable selection using Variable Selection Using Random Forest (VSURF). The number of selected features differed between different types of indices. For the Shannon index, the ideal number of variables was five, while four variables were selected for the Simpson index and for species richness (Richness) (Table 4 and Figure 4).

The variable selection method revealed that, among the 15 candidate variables, two of them were selected for all models. FHD was the GEDI variable selected for all diversity models; RDVI was the spectral index from Sentinel-2 also selected in all models. Additionally, the variable selection method for the Shannon and Simpson models revealed that GEDI metrics were more important in relation to other data sources. For these models, FHD and RH98 were selected to estimate species diversity in the Cerrado. This shows that, for the Simpson and Shannon models, GEDI variables represented 50% and 40% of the total selected variables, respectively. The auxiliary variables, Elevation (SRTM) and Precipitation (ERA-5), also proved to be important for diversity models, as they were selected in two of the three models developed (Shannon and Richness).

In general, topographic variables also proved to be relevant for variable selection. All models included at least one topographic variable. Terrain elevation was selected for the Shannon and Richness models, while terrain slope proved to be important for the Simpson model. For the 5 Sentinel-2 candidate spectral index variables, only RDVI was selected for the species diversity estimation in Cerrado. RDVI was selected for all models. The OOB error values decreased continuously as the variable number increased from 0 to 4 for Simpson and Richness models, and increased from 0 to 5 for the Shannon model.

### 3.2. Predictive Models for Species Diversity Indices

Overall, all models performed well during training with R^2^ > 0.858, RMSE < 45%, bias < −7.94 (Figure 5). Among the three models, Richness showed the best model fit with an R^2^ of 0.89; Shannon exhibited slightly lower performance with an R^2^ of 0.87; Simpson demonstrated the lowest performance with an R^2^ of 0.85. The Shannon and Richness indices were more accurately estimated with models, producing R^2^ values of 0.52 and 0.56, respectively, and an RMSE of 36% and 54% in validation, respectively (Table 5). On the other hand, the model that estimates the Simpson index showed low performance during validation (R^2^ = 0.24), despite the RMSE percentage of this model in validation being the lowest (35.29%) relative to the others.

The models tended to overestimate the values for diversity indices in the Cerrado. The species richness model produced an RMSE of 5.03; Shannon and Simpson produced RMSEs of 0.63 and 0.24 in validation, respectively. However, all models produced low bias values. Shannon and Simpson both had a bias value of 0.07; Richness had a value of –0.56.

### 3.3. Diversity Index Characterization Across the Cerrado Biome

Species diversity indices estimates were obtained by applying the models to the in-orbit GEDI data stacked with optical data (Sentinel-2, ERA-5, and SRTM). Estimates were obtained for the entire Cerrado biome using GEDI footprints with a radius of 25 m. The spatial variation of estimates of species diversity indices in the Cerrado is shown in Figure 6. These maps allowed us to identify regions in the Cerrado with greater species diversity (e.g., ~45° W ~ 5° S Figure 6(a1)) and locations with lower diversity (e.g., ~47°W ~ 16°S Figure 6(a1)). The distribution of estimates were primarily evenly distributed for all indices (Figure 6). The estimated mean values of Shannon, Simpson, and Richness were 0.99 ± 0.38, 0.37 ± 0.10, and 08.37 ± 2.90, respectively. The uncertainty of the predictions was evenly distributed across the Cerrado (Figure 7), with a pattern of lower uncertainty in regions with more GEDI footprints (Figure 7(a2–c2)).

## 4. Discussion

In this study, we were able to estimate large-scale forest diversity for the Brazilian tropical savanna (Cerrado), using GEDI data combined with conventional passive optical imagery from space. This study represents a first step towards understanding the relationship between tree species diversity and the variability of multi-source remote sensing data in the Cerrado. To date, no similar studies have been carried out in this ecosystem, and this study suggests promising potential in using free satellite-derived structural and optical data in combination with machine learning to map tree species diversity across the Cerrado.

The spatial predictions of tree species diversity are scarce for savanna ecosystems. Our results have demonstrated that the combined use of GEDI and conventional passive optical imagery data can improve large-scale species diversity indices estimates. According to [29], descriptors of alpha diversity such as Shannon, which is less affected by rare species than Richness, would be more readily predicted by remote sensing data. This aligns with our findings; however, Shannon and Richness were more accurate than Simpson. The Simpson index gives more weight to species with higher proportions whereas Richness is only based on species presence/absence. The addition of GEDI variables appears to have contributed significantly to species richness estimates, as the presence of rare species in the understory influences species richness, which might be challenging to detect using only passive sensors. Conversely, dominant species are more easily identified. These findings suggest that multi-source models that combine structural and optical data can be used to map large-scale tree species diversity in the Cerrado. The greater precision of the Shannon and Richness indices, compared to Simpson, suggests that the models are better suited for mapping diversity on large scales, especially in complex ecosystems like the Cerrado. This could guide conservation efforts in priority regions, maximizing the efficiency of available resources. Moreover, the consistent performance of the indices indicates that GEDI- and optical imagery-based models can support conservation policies by reliably identifying biodiversity hotspots in the Cerrado.

Knowledge of the variables that contributed most to model accuracies is important in modeling. It helps to select key variables that are robust and it reduces redundancy and noise in the prediction and characterization of vegetation attributes [86]. Regarding the variables selected by the VSURF algorithm, Foliage Height Diversity (FHD) proved to be very important for all diversity models. FHD emerged as the most significant variable for Simpson and the second most influential for Shannon and Richness, likely due to its capacity to indicate more complex forest structures. This characteristic is particularly relevant for ecosystems like savannas, where structural complexity varies considerably. Structural differences between tree species provide a different directional gap probability, which underlies LiDAR-based forest diversity estimates, and have been confirmed by direct correlations between tree species diversity by indices and FHD derived from GEDI. RH98 also proved to be an important variable for estimating large-scale diversity indices in the Cerrado. Canopy height information has been evaluated as an important variable for modeling forest parameters, such as biomass [53,86], fuel load [50], and forest volume [87], even in studies that estimated species diversity [88,89]. GEDI metrics are thus poised to become essential for accurately estimating tree species diversity across extensive regions in the Cerrado. The role of metrics like FHD and RH98 indicates that LiDAR data can be expanded to monitor the structural dynamics of vegetation in other savanna-like biomes. This insight paves the way for developing more robust models that incorporate seasonality and climatic patterns. Additionally, the vertical structure diversity captured by GEDI may also contribute to fauna diversity, providing greater diversity of habitats (e.g., open fields, tree canopies, and under canopies).

The Random Forest algorithm, although robust for capturing nonlinear relationships and handling imbalanced data, has limitations. These include a tendency to overfit, particularly with small datasets like the one used in this study, and a lack of transparency in the decision-making process, which makes it difficult to identify the key variables driving predictions. To address these issues, we employed cross-validation methods and variable importance analysis to enhance the reliability and interpretability of the results. Performance metrics such as R^2^, RMSE, and bias reveal both the strengths and limitations of the model in predicting species diversity in the Cerrado. The Shannon index (R^2^ = 0.52) demonstrated a moderate ability to explain variability in diversity, while the Richness index (R^2^ = 0.56) showed slightly better results due to its simplicity in quantifying the total number of species. On the other hand, the Simpson index (R^2^ = 0.24) exhibited low predictive capacity, possibly due to that model’s reduced sensitivity to patterns of uniformity in highly heterogeneous environmental areas. High RMSE values, particularly for Shannon (0.63, 37.47%) and Richness (5.03, 53.89%), indicate significant errors that may undermine practical applications. Additionally, bias analysis revealed tendencies toward overestimation for Shannon (4.44%) and Simpson (10.2%), and underestimation for Richness (−6.02%), highlighting the need to recalibrate the models to reduce biases and improve predictive accuracy, especially in biodiversity conservation and management contexts.

Our findings also showed the importance of the spectral information revealed here by RDVI from the Sentinel-2. The RDVI was strongly associated with all tree species diversity indices derived from field data. RDVI uniquely combines the advantages of NDVI and the Difference Vegetation Index (DVI), making it effective across a wide range of vegetation densities, from sparse canopies to dense forests [90]. All vegetation indices tested in this study cover the region of the infrared and red electromagnetic spectrum, which are the wavelength regions often defined as the most relevant for studying differences in vegetation structures [91,92]. Previous studies have found that the use of VIs is better correlated with abundance indices (Shannon and Simpson) [19,93], but they can also partially explain [19]. Furthermore, [94] reported that spectral information is better used to explain variations in forest structures at lower levels of biomass. The RDVI has a measurement scale that ranges from 0 to well beyond 1, and such an open scale facilitated its ability to explain tree species diversity in our study. Most studies report NDVI as a potential variable for forest species diversity; however, the scale problem of the NDVI has long been recognized as limiting its ability to detect forest canopy variation [23] and, therefore, it is not surprising that the RDVI has greater explanatory power than the other indices.

Topographic data source was also important for the models. Elevation was important for the Shannon and Richness models; Slope was necessary for the Simpson model. The importance of elevation and slope has also been emphasized in previous studies that explored tree species classification, which reflects their role in shaping environmental conditions such as solar radiation exposure and water availability [95,96]. Previous studies claim that elevation was one of the most important metrics in estimating tree species diversity [97,98,99]. This statement aligns with the result of the Richness model, as elevation was the most important variable for this model. Our findings support the idea that elevation, combined with precipitation, significantly affects tree species distribution and diversity. These studies explored the relationship between vegetation and topographic factors and found that, with an increase in elevation and aspect, the growth rates of vegetation also increased. In addition, as reported by [100], not all diversity indices are equally correlated with landscape parameters. For example, species richness correlates better with landscape parameters than the Shannon index. This is partially in line with our results, in which elevation was important for the Richness model as well as the Shannon model. These results reinforce those from other studies where remotely derived abiotic factors related to topographic and edaphic properties were found to be significant predictors of tree species richness [101,102,103]. Additionally, precipitation, another critical abiotic variable, was integral to the Shannon and Richness models, reinforcing its role as a key driver of biodiversity patterns in tropical ecosystems. This suggests that remote sensing explains variations in species diversity better when integrated with environmental variables, given that they are also known to influence the spatial patterns of natural resources [104,105]. Areas with higher species diversity coincided with regions of greater topographic heterogeneity, suggesting that structural complexity is crucial for maintaining biodiversity. However, high-diversity regions in non-protected areas face significant risks from predatory exploitation and agricultural conversion, underscoring the need for expanding protected areas and creating ecological corridors to connect existing reserves and to mitigate habitat fragmentation.

Although our models have proven to be consistent for estimating and mapping large-scale species diversity indices in the Cerrado, this study has some limitations, and the methodology can be improved in the future. First, diversity indices were not generated separately for different zone types (communal and protection), as described in Section 2.1. Such an analysis could have provided more detailed insights into the differences in diversity patterns, based on management and conservation policies. Second, the analysis focused exclusively on tree species, excluding understory and herbaceous strata, which play a critical role in overall biodiversity in the Cerrado. Including these strata in future studies would broaden our understanding of diversity patterns in this biome. Lastly, the study did not explore beta and gamma diversity, which are essential in understanding species variation across areas and total landscape diversity. Incorporating these analyses in future work could offer a more comprehensive perspective on biodiversity and its spatial distribution. In addition, the use of a small number of plots (50), as well as logistical challenges in covering all vegetation types in the Cerrado, impacted the robustness of the model. While GEDI data capture biodiversity patterns in forest ecosystems without ancillary data, its applicability to savanna ecosystems like the Cerrado is still uncertain, which highlights the need for more comprehensive data. Future research should incorporate multi-temporal datasets to account for seasonal variations, as demonstrated in savanna and forest studies, where the timing of analyses significantly influenced diversity modeling. In this study, median composites from May and August ensured cloud-free data but focused on the dry season. Spectral resolution also proved essential, with additional wavelengths, such as the red-edge band, improving the accuracy of diversity estimates. Finally, improving plot design, e.g., by using clusters, could better capture landscape variability. Despite these limitations, the methods developed can be applied to other tropical and savanna biomes, contributing to global biodiversity assessments and studies on climate change. Despite these limitations, the results provide an important foundation for understanding species diversity in the Cerrado and for developing conservation and management strategies. Future studies could integrate broader approaches, such as including multiple vegetation strata and more detailed analyses according to management zones, and by applying methods to assess beta and gamma diversity.

Habitat heterogeneity has often been associated with species richness. Several authors have reported strong associations between spectral heterogeneity (as a proxy for habitat heterogeneity) and species richness [17,106]. Future studies could be concerned with forming a more robust database by increasing the number of plots in each vegetation type, in addition to analyzing different regions. Furthermore, future research may address whether there are significant differences in model predictions caused by seasonality, given that the results of research carried out in forest formations that belong to distant biomes and are subject to different climate regimes can be quite contrasting [107]. It would also be interesting to use new approaches to increase the quantity and coverage of canopy structure measurements by GEDI, as the mission was programmed to collect only traces. Future studies should also focus on other multi-spectral vegetation diversity indices, including those extracted from red-edge bands, as they enable the recording of subtle differences in leaf structure and chlorophyll content [108,109].

Statistically rigorous methods for communicating results quickly to the scientific community, government, industrial stakeholders, and the general public are a critical element of successful biodiversity integrity monitoring programs [110,111,112,113]. Understanding tree species diversity status in biodiversity conservation is crucial, as it provides management with the necessary baseline information about tree species distribution in a given ecosystem, which is essential in planning and management. This type of information is of great importance for the Cerrado, which has been subject to intense predatory exploitation: countless animals and plants are at risk of extinction, and only 8.21% of the total area of the territory is legally protected by conservation units. As suggested by [32], we also believe that future research should focus on the transferability of structure-diversity models to other regions and continents in order to more precisely establish the potential of this method in various biomes. Topics such as the assessment of biodiversity loss in ecosystems are extremely important and could be better explored. The information derived from our approach may open new opportunities for future studies of global importance.

## 5. Conclusions

We developed a new workflow for large-scale tree species diversity mapping for the Brazilian tropical savanna (Cerrado) combining GEDI, optical, and environmental data. Three multi-source models were developed to estimate large-scale tree species diversity indices in the Cerrado. The Shannon and Richness indices were more consistent than the Simpson index in evaluating tree species diversity in the Cerrado. GEDI metrics can capture information related to vegetation structures and significantly improve the accuracy of tree species diversity estimates and mapping. The RDVI vegetation index was selected for all models, confirming our hypothesis that spectral information contributes to the description of species diversity. The elevation and slope variables were also necessary for composing the models; precipitation was more favorable for predicting the Shannon and Richness indices. Remote sensing provides systematic spatial and temporal data on vegetation attributes, which can be assessed on a large scale. Our findings will contribute to the identification of locations with high and low species richness, and an abundance of shrubs and trees, through remote sensing data. The maps provided in this study will be valuable for the assessment and management of tree species diversity in the Cerrado. Given the Cerrado’s status as a biodiversity hotspot, as well as its vulnerability to anthropogenic pressures, our findings underscore the importance of leveraging advanced technologies like GEDI to ensure its sustainable management and conservation.

## Figures and Tables

**Figure 1 sensors-25-00308-f001:**
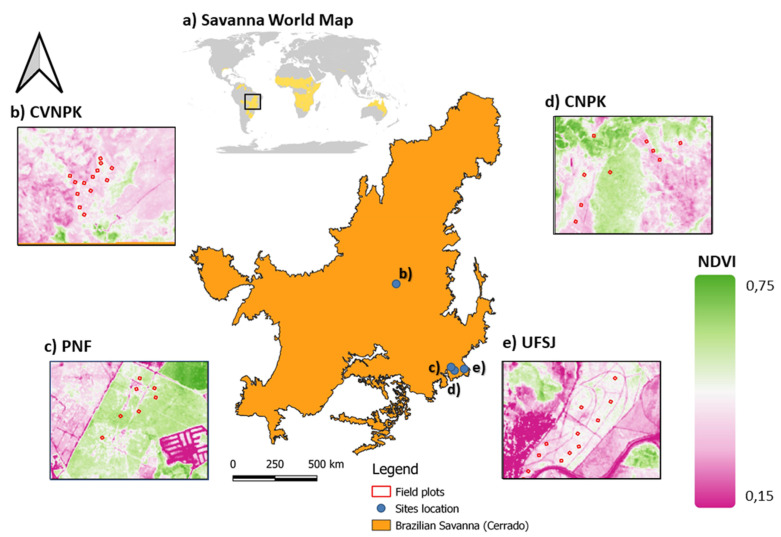
(**a**) Spatial location of the Brazilian savanna (Cerrado) and study sites where UAV-lidar and field data were collected: (**b**) Chapada dos Veadeiros National Park (CVNPK); (**c**) Paraopeba National Forest (PNF); (**d**) Serra do Cipó National Park (CNPK); (**e**) University of São João Del-Rei’s Forest (UFSJ).

**Figure 2 sensors-25-00308-f002:**
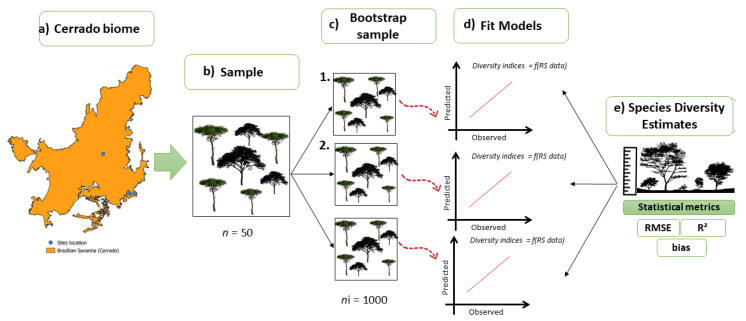
Flowchart presenting the bootstrapping method for modeling diversity indices in the Brazilian Cerrado: (**a**) Cerrado and field plots; (**b**) area sampling (total 50 plots); (**c**) from the sample-bootstrap method with a total of 1000 repetitions; (**d**) fit models developed after bootstrap sampling for each diversity index; (**e**) validation of fitted models and statistical metrics for each diversity model.

**Figure 3 sensors-25-00308-f003:**
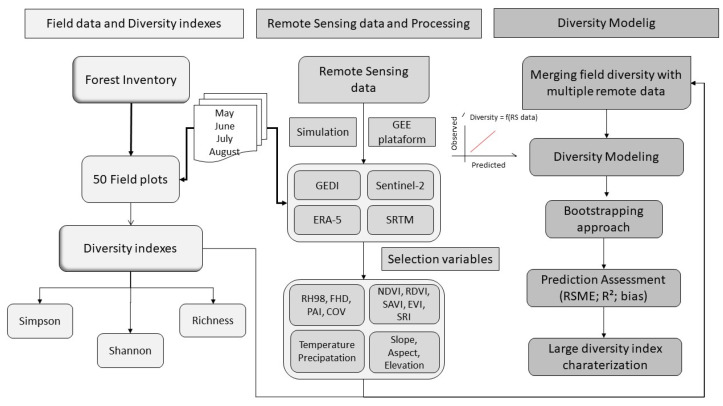
Methodological flowchart for processing multi-source remote sensing data and modeling diversity in different landscapes in the Cerrado.

**Figure 4 sensors-25-00308-f004:**
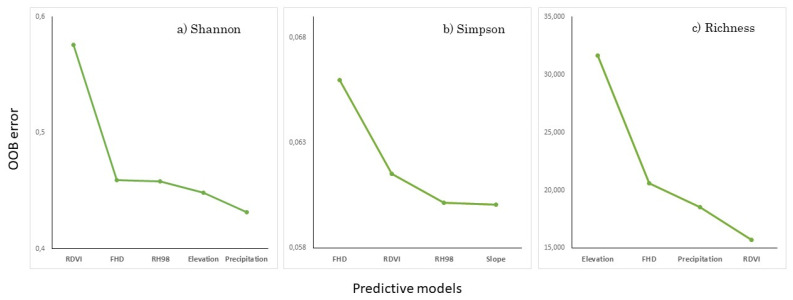
Variable Selection Using Random Forest (VSURF), giving the number of variables that meet the requirements for prediction.

**Figure 5 sensors-25-00308-f005:**
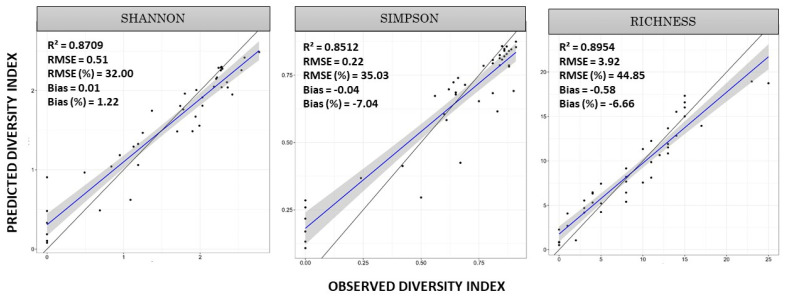
Training results for estimating the diversity indices (Shannon, Simpson, and Richness), using Random Forest, GEDI waveform metrics, and passive optical imaging as predictors. R^2^ = coefficient of determination; RMSE = root mean square error; and bias.

**Figure 6 sensors-25-00308-f006:**
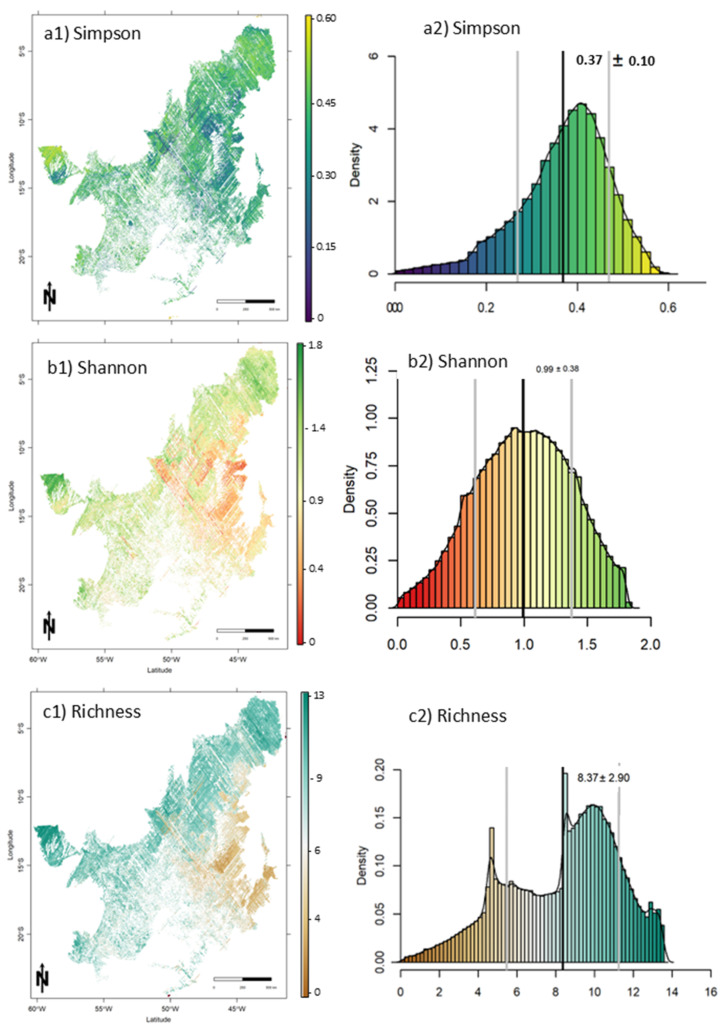
Large-scale diversity indices prediction maps (**a1**–**c1**) and distribution (**a2**–**c2**) at 1 km spatial resolution for the entire Cerrado biome.

**Figure 7 sensors-25-00308-f007:**
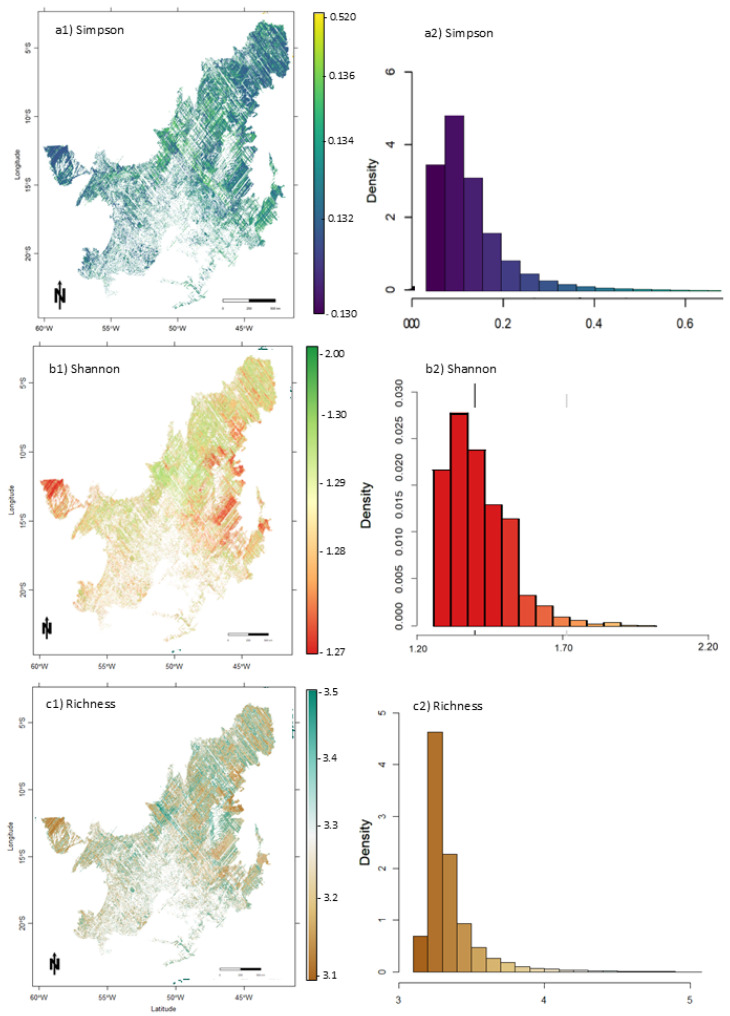
Large-scale diversity indices uncertainty prediction maps (**a1**–**c1**) and distribution (**a2**–**c2**) at 1 km spatial resolution for the entire Cerrado biome.

**Table 1 sensors-25-00308-t001:** Diversity indices used in the study and their equations, where *pi* is the percent cover proportion of the species.

Diversity Metric	Equation
Species richness (S)	Number of species
Shannon’s index (H′)	H′ = ∑ pi Ln pi
Simpson’s index (D)	D = ∑ pi^2^

**Table 2 sensors-25-00308-t002:** Vegetation indices derived from images from the Sentinel-2 satellite MSI sensor.

Indices	Equation
Normalized Difference Vegetation Index	NDVI = NIR − RED/NIR + RED
Renormalized Difference Vegetation Index	RDVI = NIR − RED/√NIR + RED
Soil-Adjusted Vegetation Index	SAVI = NIR − REDNIR + RED + 0.5 × 1 + 0.5
Enhanced Vegetation Index	EVI = 2.5 × NIR − REDNIR + 6 × RED − 7.5 × A + 1
Simple Ratio	SR = NIR/RED

**Table 3 sensors-25-00308-t003:** Sets of remote sensing candidate metrics for the forest species diversity modeling.

Metric Set	Source Name	Variables
1	GEDI	RH98 + FHD + PAI + COV
2	SENTINEL 2	NDVI + SR + SAVI + RDVI + EVI
3	ERA 5	Temperature + Precipitation
4	SRTM	Slope + Aspect + Elevation

**Table 4 sensors-25-00308-t004:** Variable Selection Using Random Forest (VSURF) for each diversity index model.

Diversity Index	Selected Variables
Shannon	RDVI + FHD + RH98 + Elevation + Precipitation
Simpson	FHD + RDVI + RH98 + Slope
Richness	Elevation + FHD + Precipitation + RDVI

**Table 5 sensors-25-00308-t005:** Cross-validation performance assessment in 500 iterations of models used to estimate the diversity index (Shannon, Simpson, and Richness), using Random Forest, GEDI waveform metrics, and conventional passive optical imaging as predictors. R^2^ = coefficient of determination; RMSE = root mean square error; bias; and t.

Diversity Index	R^2^	RMSE	RMSE%	Bias	Bias%
Shannon	0.52	0.63	37.47%	0.07	4.44
Simpson	0.24	0.24	35.29%	0.07	10.2
Richness	0.56	5.03	53.89%	−0.56	−6.02

## Data Availability

The data used in this study are available upon request from the corresponding author. Restrictions apply to the availability of certain data due to [e.g., privacy or legal reasons].

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
