# Peer review of "Spatial Characterization of Woody Species Diversity in Tropical Savannas Using GEDI and Optical Data"

_sensors, 2025, doi:10.3390/s25020308_

Round 1

Reviewer 1 Report

Comments and Suggestions for Authors

1. The manuscript includes an excessive number of references. It is recommended that the authors condense the summary of the current research status and omit unnecessary references to maintain focus and improve clarity.

2. Some references are incomplete, such as references 53 and 60. Please verify the content and formatting of all references to ensure consistency and accuracy.

3. In line 196, the manuscript states that 50 square plots were measured between June and July of 2010, while GEDI began collecting scientific data in March 2019. The significant time gap between these datasets raises concerns about the potential impact on the research results. Please elaborate on how this time difference might influence the findings.

4. In lines 156 and 192, and Figure 1 (d), different abbreviations—SCNP, SCNPK, and CNPK—are used to refer to the same study area, Serra do Cipó National Park. It is strongly recommended to adopt a unified abbreviation throughout the manuscript for consistency and clarity.

5. Sections 2.2 and 2.3 share the same title, Field and forest diversity data, while Sections 2.3.1 and 2.3.2 both use the title UAV-LiDAR GatorEye, discussing the simulation of GEDI data. Consider merging the content of these sections to avoid redundancy and streamline the structure.

6. To enhance the clarity of the prediction and distribution maps presented in Figures 6 and 7, it is advisable to incorporate legends for both the species diversity index prediction and distribution maps (Figure 6) as well as the uncertainty prediction and distribution maps (Figure 7). This addition will facilitate a more accurate interpretation of the results by readers.

7. All tables within the manuscript are labeled generically as 'This is a table.' It is essential that each table be assigned a specific title pertinent to its research content, enabling readers to intuitively grasp its purpose and findings.

Author Response

  1. Excessive number of references in the manuscript
    Thank you for highlighting this issue. We revised the manuscript to condense the summary of the current research status, retaining only the most essential and relevant references. 
  1. Incomplete references
    We acknowledged that references 53 and 60 were incomplete. A thorough review of all references cited in the manuscript was conducted, ensuring they are now complete, properly formatted, and consistent with the citation style used.
  1. Temporal gap between plot measurements and GEDI data
    We appreciate your observation regarding the time gap. Upon review, we identified that this was a typographical error in the manuscript. The correct year for the field measurements is 2019, not 2010. Therefore, there is no temporal gap between the field data and the GEDI data. The manuscript has been updated to reflect this correction.
  1. Inconsistent abbreviations for Serra do Cipó National Park
    The inconsistent use of SCNP, SCNPK, and CNPK for Serra do Cipó National Park was corrected. The manuscript was revised to adopt a single abbreviation, "SCNP," ensuring consistency and improving readability.
  1. Redundant titles in Sections 2.2 and 2.3, and subsections 2.3.1 and 2.3.2
    The redundancy in the titles of Sections 2.2 and 2.3, as well as subsections 2.3.1 and 2.3.2, was resolved. The content was consolidated, and the titles were reorganized to accurately reflect the topics discussed. This adjustment streamlined the structure and eliminated repetition.
  1. Missing legends in Figures 6 and 7
    To facilitate the comprehension of the results presented in Figures 6 and 7, we have restructured the Results and Discussion sections. The alterations include a detailed revision of the text that describes the prediction and distribution maps, providing a clearer and more precise interpretation of the data.
  2. Generic table titles
    The generic titles for all tables were replaced with specific titles that reflect the content and purpose of each table. 

If further clarification or additional adjustments are required, I am happy to address them.

Reviewer 2 Report

Comments and Suggestions for Authors
  1. Clarity and Organization of the Paper:
    The paper is generally well-organized, but there are some sections that could benefit from further clarification. For instance, the Introduction could provide a more concise overview of the research problem and the motivation behind using GEDI and optical data to study species diversity in tropical savannas. Additionally, the Materials and Methods section could be more detailed in explaining the specific steps taken in data collection and processing.

  2. Rigor of Methodology:
    The methodology employed in the study appears to be rigorous, combining multiple data sources and using advanced statistical models. However, it would be helpful to include more information on the validation of the simulated GEDI data and the uncertainty associated with these data. Furthermore, a discussion on the potential limitations of the Random Forest models used in the study would be beneficial.

  3. Originality and Contribution to Knowledge:
    The paper makes a valuable contribution to the field by exploring the use of GEDI and optical data to predict species diversity in tropical savannas. However, the authors could emphasize more clearly how their study adds to the existing body of knowledge and what new insights it provides. Additionally, a comparison with other similar studies would help to highlight the originality of the research.

  4. Presentation of Results:
    The results section is well-presented, with clear tables and figures illustrating the findings. However, some additional analysis or interpretation of the results would be helpful. For example, a discussion on the implications of the performance metrics (R², RMSE, and Bias) for the practical application of the models would be beneficial. Additionally, the authors could provide more insights into the spatial patterns of species diversity observed in the Cerrado biome.

  5. Readability and Language Use:
    The paper is generally well-written and easy to follow. However, there are some instances of technical jargon and abbreviations that may be unfamiliar to readers outside the field. The authors could consider defining these terms or providing a glossary to improve readability. Additionally, some minor editing and proofreading would be helpful to correct any grammatical or typographical errors.

Author Response

  1. Clarity and Organization of the Paper
    We acknowledge the need to improve the clarity and organization of the paper. The introduction has been revised to provide a more concise overview of the research problem and the motivation for using GEDI and optical data to study species diversity in tropical savannas. Additionally, the Materials and Methods section has been expanded to include a clearer description of the specific steps involved in data collection and processing.

  1. Rigor of Methodology

    Thank you for your insightful comments. We appreciate the opportunity to clarify our methodology and address your concerns.

    Regarding the validation of the simulated GEDI data, we relied on the findings from The GEDI simulator: A large-footprint waveform lidar simulator for calibration and validation of spaceborne missions by Hancock et al. (2019). This study demonstrates that the GEDI simulator produces highly accurate waveform metrics when compared to observed large-footprint, full-waveform lidar data from NASA's LVIS. Specifically, the validation showed a mean bias of less than 0.22 m and a root mean square error (RMSE) of less than 5.7 m, provided that the airborne laser scanning (ALS) data have sufficient pulse density. Additionally, measurement errors due to instrument noise were within 1.5 m of observed waveforms, with 70–85% of variance in measurement error explained by the simulator. These results underscore the robustness of the GEDI simulator, giving us confidence in its ability to produce reliable data for our study.

    We recognize the importance of discussing the potential limitations of the Random Forest models used in the study. While Random Forests are robust and versatile, they can be prone to overfitting, especially when handling noisy or imbalanced data. We have revised the manuscript to include a discussion on these limitations and the steps taken to mitigate them, such as cross-validation and feature selection strategies. We hope this addresses your concerns and enhances the clarity of our methodology. Thank you for your valuable feedback, which has strengthened the rigor of our study.

  1. Originality and Contribution to Knowledge
    The originality of the study has been explicitly emphasized. The text has been revised to highlight how our research contributes to the existing body of knowledge and what new insights it provides. Additionally, a comparison with similar studies has been included to underscore the uniqueness of the approach adopted.

  1. Presentation of Results
    We recognize the importance of a more in-depth analysis of the results. The discussion has been expanded to address the implications of the performance metrics (R², RMSE, and Bias) for the practical application of the models. Moreover, we have provided a more detailed analysis of the spatial patterns of species diversity observed in the Cerrado biome.

  1. Readability and Language Use
    The text has been carefully revised to clarify technical terms and abbreviations that may be unfamiliar to readers outside the field. Where necessary, definitions have been added. Additionally, the manuscript has undergone editing to correct any grammatical or typographical errors, improving the overall readability and flow.

Should further adjustments or clarifications be required, we remain available to address them.

Reviewer 3 Report

Comments and Suggestions for Authors

Observations and suggestions have been made in the manuscript to improve its content and form. In each section of the manuscript, specific observations have been indicated that the authors must address.

In Introduction: In this section, bibliographic citations that support various arguments must be included. Quantitative values should also be included to give an idea of how the "large areas" should be conceived.

In Materials and Methods: This section must complement relevant quantitative information and include specific technical details. In addition, bibliographic citations that support various methodological aspects must be included. The tables in this section are missing their respective titles, please include them.

In Results: In this section, the mathematical structure of the final models obtained for each diversity index must be presented in full. All the parameters that make up each model must be included, as well as the respective goodness-of-fit statistics.

In Discussion: This section must include the logical arguments and causes that explain the results that were indicated in the Results section. Regarding the limitations of the study, the following must be included: (1) There was a lack of generating diversity indices by type of zone (communal and protection) of the study area that the authors indicate in section 2.1.; (2) That only the diversity in tree species was explored, the analysis of diversity indices for the understory and the herbaceous stratum is missing; (3) Beta and gamma diversity were not analyzed.

In References: In this section, many inconsistencies are observed in the way each bibliographic reference is written. Only a few are noted. The authors must review the writing of this section and correct where appropriate. The style of the Journal must be respected.

Author Response

Introduction
Thank you for the insightful suggestions. We have restructured the introduction to provide a clearer and more concise overview of the research problem and motivation, as suggested. This revised structure now emphasizes the use of GEDI and optical data to study species diversity in tropical savannas, aligning with the feedback received.

Materials and Methods
We greatly appreciate your observations. Adjustments and changes have been made to this section, and additional details have been included to elaborate on steps that were previously oversimplified. Titles have been assigned to all tables in this section to clarify their content and purpose, ensuring better understanding by the readers.

Results
The structure of the final models obtained for each diversity index is now presented in Table 4, and the statistics of each model are shown immediately afterward. Furthermore, we expanded the discussion of the results, particularly in this section, to enhance comprehension for readers, providing a more thorough analysis of the implications.

Discussion
We adhered to the excellent contributions suggested for this section and have incorporated all the points raised. Logical arguments and detailed explanations for the results have been added, along with discussions on the study’s limitations and the aspects that were not addressed (e.g., diversity indices by type of zone, understory and herbaceous strata diversity, beta and gamma diversity). We are grateful for these suggestions, which significantly enriched the manuscript.

References
We thank the reviewers for identifying inconsistencies in the References section. A thorough review has been conducted, and corrections have been made to ensure consistency and alignment with the journal's style. All references now comply with the required format.

We sincerely appreciate the constructive feedback, which greatly contributed to the improvement of the manuscript. Should there be any additional comments or questions, we remain available to address them. 

Round 2

Reviewer 1 Report

Comments and Suggestions for Authors

After thorough revision, the manuscript has largely satisfied the journal's requirements. Additionally, I have two minor suggestions to further enhance the paper prior to publication.

1It is recommended to incorporate a schematic diagram illustrating the GEDI working mode in section 2.3.2, as this will facilitate a more comprehensive understanding of the content for the readers.

2In section 2.4, the development of predictive models and validation should be described in detail to enable readers to comprehend the data flow, thereby assessing the accuracy and feasibility of the model. How are the GEDI and optical data used together?

Author Response

Comments 1: 

It is recommended to incorporate a schematic diagram illustrating the GEDI working mode in section 2.3.2, as this will facilitate a more comprehensive understanding of the content for the readers.

Response 1: 

While we acknowledge the suggestion to include a schematic diagram illustrating the GEDI working mode in section 2.3.2, we believe this addition may not be necessary in this context. The primary focus of the section is on the integration of GEDI data with optical and ancillary datasets for predictive modeling of species diversity. The core methodology already highlights how the GEDI Level 2A and 2B products are utilized, filtered, and processed within the workflow.

Furthermore, the GEDI mission and its working mode have been extensively documented in prior studies and technical resources (e.g., Dubayah et al., 2021; Silva et al., 2020), which are already cited in the manuscript. Including a schematic diagram here would duplicate existing material that is readily available to readers familiar with remote sensing technologies.

Given the focus of this study on the application of GEDI data rather than on the technical details of the instrument itself, we believe that incorporating such a diagram could detract from the manuscript’s central objective: presenting the methodology and outcomes of the diversity index models. For readers seeking further understanding of the GEDI system's operation, we have provided appropriate references to guide them to comprehensive resources.

Comments 2: In section 2.4, the development of predictive models and validation should be described in detail to enable readers to comprehend the data flow, thereby assessing the accuracy and feasibility of the model. How are the GEDI and optical data used together?

Response 2: 

Thank you for the feedback regarding section 2.4. Based on your suggestion, I have made substantial improvements to the section, providing a more detailed description of the predictive model development and the data flow. These revisions aim to ensure that readers clearly understand the integration of GEDI data with optical (Sentinel-2), climatic (ERA-5), and elevation (SRTM) datasets, as well as how these combined variables were used to develop the models.

Additionally, the title of the section has been adjusted to 2.5, as it was previously repeated from the preceding topic. These modifications enhance the clarity and structure of the text, fully addressing the comment.